# Gradients in signal complexity of sleep-wake intracerebral EEG

**Giridhar Kalamangalam**[1,2*], **Ioan Mircea Chelaru**[2], **Abbas Babajani-Feremi**[1,3]

**1** Department of Neurology, UF McKnight Brain Institute, Gainesville, Florida, United States of America, **2** Wilder Center for Epilepsy Research, University of Florida, Gainesville, Florida, United States of America, **3** Magnetoencephalography Lab, Norman Fixel Institute for Neurological Diseases University of Florida, Gainesville, Florida, United States of America

\* gkalamangalam@ufl.edu

## Abstract

Spatial variation in the morphology of the electroencephalogram (EEG) over the head is classically described. Ultimately, location-dependent variation in EEG must arise from the cytoarchitectural and network structure of the portion of cortex sensed. In previous work, we demonstrated that over the lateral frontal lobe, sample entropy (SE) of intracerebral EEG (iEEG) over a subdural recording contact was predictive of that contact's connectivity to other contacts. In this work, we used a publicly available repository (the Montreal Neurological Institute Atlas; MNIA) of whole-brain normative iEEG to calculate SE over the entire cortical surface. SE was averaged region-wise and classified by the state of arousal (awake, N2, N3 and REM). SE averages were transformed to a linear scale between zero and unity, mapped to continuous color scale and overlaid on segmented cortical surface models, one for each sleep-wake state. Wake SE followed a rostro-caudal gradient (RCG), with high values anteriorly and a global minimum in the posterior cortex. Superimposed on the RCG were other gradients radiating away from primary somatic sensorimotor, visual and auditory regions to their association areas. All gradients were attenuated in deep (N3) sleep. In REM, the majority of the cortex exhibited wake-like SE, with the prominent exception of primary cortical sensory and motor areas. Normative human intracerebral EEG exhibits rich spatial structure - cortical gradients - in the distribution of SE. SE in the wake state tracks temporal processing hierarchies in cerebral cortex, concordant to the distribution of several other cortical attributes of structure (e.g., cortical thickness, myelin content). Sleep disrupts these gradients, with REM sleep bringing out unusual discordances between primary sensory and their association areas. Our results deepen the interpretation of EEG from conventional descriptors such as Berger bands to a spatial perspective related to cortical biology.

## Introduction

Regional differences in the visual appearance of the scalp electroencephalogram (EEG) appeared in Hans Berger's original writings [1]. The systematic EEG changes accompanying sleep-wake states were discovered not long afterwards [2]. The relationship of local EEG morphology to that brain region's structure and function – for instance, connectivity profile,

**Data availability statement:** https://mni-open-ieegatlas.research.mcgill.ca

**Funding:** The corresponding author (GK) acknowledges support from the National Institute of Neurological Disorders and Stroke (R21NS128503). GK and IMC acknowledge support from the Wilder family endowments to the University of Florida. The funders had no role in study design, data collection and analysis, decision to publish, or preparation of the manuscript.

**Competing interests:** The authors have declared that no competing interests exist

cytoarchitecture, cortical thickness, myelin content, and gene expression – is however a subject of continuing research interest. For neurologists, the region-specific behavior of seizures and their relationship to baseline brain sleep-wake rhythms are of fundamental importance.

In recent work with human epilepsy patients undergoing invasive evaluation with subdurally-placed electrodes, we examined sample entropy (SE; a metric of signal complexity) of the local intracerebral EEG (iEEG). We found a reciprocal relationship between SE of a recording contact and its average connectivity to other electrodes in the neighborhood. We also found a rostro-caudal gradient (RCG) to this reciprocal relationship, with larger values of SE occurring anteriorly. The largest values of SE of all occurred over the most posterior lateral frontal brain region sampled, the primary motor cortex. We interpreted our results in terms of the functional differences between the primary motor (posterior frontal) and association (anterior frontal) cortex.

In this work, we use a publicly available repository (the Montreal Neurological Institute Atlas; MNIA) of whole-brain normative iEEG to extend SE calculations to the entire cortical surface. We were interested in whole-brain gradients of SE, and their relationship to documented gradients of other cortical attributes, and the region-specific changes accompanying sleep and wake states. A specific question of interest was whether the maximum value of SE over primary motor cortex attained in our prior study was 'local' (i.e., just over the frontal lobe) or a 'global' (whole brain) maximum.

## Methods

### Data

We accessed the Montreal Neurological Institute Open IEEG Atlas (https://mni-open-ieegatlas.research.mcgill.ca); MNIA [3, 4]. MNIA comprises intracranial EEG data from 106 subjects from three referral centers, recorded with either stereotaxic depth (SEEG) or subdural grid electrodes. The data were judged 'normative': i.e., chosen from brain regions without pathological features, and at least 72 hours after electrode insertion (for SEEG) and one week (for subdural grids), and at least 12 hours after a convulsive seizure, six hours after focal clinical seizures and two hours after purely EEG seizures, and prior to any clinical tests involving electrical stimulation. The sleep data (sMNIA) consist of 1468 recorded segments of non-rapid eye movement (NREM) sleep phase N2, 1468 of NREM sleep phase N3 and 1012 of rapid-eye movement (REM) sleep. Wake data (wMNIA) comprise 1772 segments, recorded during quiet wakefulness with eyes closed. All recordings (*w*- and *s*-MNIA) last 68s and are digitized at 200 Hz or more. All patients in MNIA contributed data to more than one location, and all sampled brain regions were contributed to by multiple subjects.

All data processing in this manuscript utilized custom software written in MATLAB® (The Mathworks, Natick, MA).

The human data used in the manuscript were initially accessed from the cited publicly available database on May 5–6, 2022. There are no relevant institutional review or ethics declarations relevant to the authors of this manuscript or their institution. The authors do not have access to information that could identify individual participants during or after data access.

### Electrode sample entropy

Sample entropy (SE) is a measure of signal richness or complexity [5]. Large values imply greater randomness, information content, or irregularity; small values conversely imply a greater presence of patterned motifs, repetitive features, or regularity in the data. SE is indexed by two parameters *m* – 'embedding dimension' - an estimate of the linear dimension of the

smallest feature of interest, and $r$ – 'tolerance' – the allowable error between similar features. For a data series of length $N$, $x = \{x_1, x_2, ..., x_N\}$ one considers a template vector of length $m$, $m$ « $N$, $x_m(i) = \{x_i, x_{i+1}, ..., x_{i+m-1}\}$, and the Euclidean distance $|x_m(i) - x_m(j)|$ $(i \neq j)$, defining

$$SE = -\ln(A / B),$$

where $A$ is the number of template vector pairs having $|x_{m+1}(i) - x_{m+1}(j)| < r$ and $B$ is the number of template vector pairs having $|x_m(i) - x_m(j)| < r$.

We used the conventional choices of $m = 2$ and $r = 0.2$ [6], verifying that these provided the best statistical separation for the data (below) in comparison to a range of other choices ($m = 3, 4, 5; r = 0.1, 0.3, 0.4, 0.5$). Each 68s raw data record was normalized prior to computing SE in moving windows of 5s length that overlapped for 4s. Raw SE values were normalized into the range [0,1] using the global maximum and minimum values. The mean normalized SE and its standard deviation for a particular location were computed from all the available samples from that brain region, and separately for each sleep-wake stage.

### Region and state-specific SE

The mean normalized SE values for each state (wakefulness, N2 sleep, N3 sleep, and REM sleep) for each of the 38 MNIA brain regions are presented in Table 1. Table 2 lists the fractional changes between the wake and each sleep state.

### Statistical analysis

The Wilcoxon rank sum test was used to test the significance of the difference between the distribution of SE values for the wake versus N2, wake versus N3 and wake versus REM, Bonferroni-corrected for multiple comparisons. Differences in spatial distribution of SE between states were examined by permuting (shuffling) the vector of SE values within individual states. Each average SE vector (the Table 1 columns of average wake, N2, N3 and REM SE values) was shuffled 10,000 times. In standard notation, the cosine similarity (CS)

$$CS_{U,V} = \frac{U \cdot V}{|U||V|}$$

of pairs ($U, V$) of shuffled vectors within that single condition was assembled into a distribution. The $t$-test was used to compare CS of the average SE between different states (i.e., wake-N2, N2-N3, wake-REM) and these distributions.

### Brain surface display

Table 1's SE values were transformed to colors with the predefined *jet* colormap in MATLAB® and overlaid on to a cortical surface model using FreeSurfer's [7] Destrieux atlas [8]. The 3-D volumetric representations of the amygdala and hippocampus were based on FreeSurfer's whole-brain automatic segmentation. The mapping of the 38 MNIA brain regions to the 74 anatomical parcellations in the Destrieux atlas and the amygdala and hippocampus was similar, though somewhat more detailed, than in our previous work [9]. The larger numbers of total Destrieux brain regions meant that there were several instances of a single MNIA region mapping to multiple Destrieux brain regions. Thus, the fine cortical parcellation of the Destrieux atlas was coarsened by its mapping from MNIA. For example, MNIA's single 'superior frontal gyrus and frontal pole' mapped to areas 1, 5, 16 and 54 of Destrieux. However, along the frontomedial brain surface the MNIA parcellation is finer than Destrieux. In the immediate peri-Rolandic area, MNIA lists three regions: the *precentral gyrus*, the *medial*

**Table 1. Normalized sample entropy values by MNIA brain region and sleep-wake state.**

| | MNIA region | SEW | SEN2 | SEN3 | SER |
|---|---|---|---|---|---|
| 1 | Superior and middle occipital gyri | 0.5174 | 0.4045 | 0.0423 | 0.5917 |
| 2 | Inferior occipital gyrus & occipital pole | 0.4081 | 0.3803 | 0.0501 | 0.4299 |
| 3 | Cuneus | 0.5593 | 0.3460 | 0.1654 | 0.5009 |
| 4 | Calcarine cortex | 0.6656 | 0.3748 | 0.1429 | 0.3871 |
| 5 | Lingual gyrus & occipital fusiform gyrus | 0.5943 | 0.3694 | 0.1013 | 0.4594 |
| 6 | Postcentral gyrus with medial segment | 0.7916 | 0.4064 | 0.0629 | 0.6875 |
| 7 | Superior parietal lobule | 0.6449 | 0.4701 | 0.0882 | 0.6667 |
| 8 | Parietal operculum | 0.7804 | 0.5097 | 0.1624 | 0.8808 |
| 9 | Supramarginal gyrus | 0.7137 | 0.4503 | 0.0965 | 0.6706 |
| 10 | Angular gyrus | 0.6453 | 0.3867 | 0.0327 | 0.6396 |
| 11 | Precuneus | 0.7061 | 0.5166 | 0.1336 | 0.7470 |
| 12 | Posterior cingulate | 0.8366 | 0.4029 | 0.1087 | 0.6014 |
| 13 | Anterior insula | 0.7544 | 0.4252 | 0.0827 | 0.6856 |
| 14 | Posterior insula | 0.7147 | 0.3953 | 0.1003 | 0.5782 |
| 15 | Gyrus rectus and orbital gyri | 0.7032 | 0.3658 | 0.0683 | 0.8071 |
| 16 | Anterior cingulate | 0.6177 | 0.3717 | 0.1203 | 0.6078 |
| 17 | Middle cingulate | 0.8311 | 0.5292 | 0.2492 | 0.8024 |
| 18 | Supplementary motor cortex | 0.8622 | 0.5242 | 0.2145 | 0.9110 |
| 19 | Medial frontal cortex | 0.7058 | 0.3159 | 0.0095 | 0.6881 |
| 20 | Central operculum | 0.7660 | 0.4481 | 0.1616 | 0.8198 |
| 21 | Frontal operculum | 0.7792 | 0.4491 | 0.0898 | 0.7555 |
| 22 | Opercular part of inferior frontal gyrus | 0.8973 | 0.5028 | 0.1848 | 0.8099 |
| 23 | Triangular part of inferior frontal gyrus | 0.7673 | 0.4658 | 0.2042 | 0.8335 |
| 24 | Orbital part of inferior frontal gyrus | 0.8122 | 0.4713 | 0.1250 | 0.8347 |
| 25 | Middle frontal gyrus | 0.8649 | 0.4704 | 0.1924 | 0.8874 |
| 26 | Superior frontal gyrus and frontal pole | 0.8168 | 0.4249 | 0.1284 | 0.9111 |
| 27 | Medial segment of superior frontal gyrus | 0.7883 | 0.3278 | 0.1411 | 0.7510 |
| 28 | Medial segment of precentral gyrus | 1.0000 | 0.5029 | 0.1397 | 0.7606 |
| 29 | Precentral gyrus | 0.9363 | 0.5558 | 0.2612 | 0.8224 |
| 30 | Superior temporal gyrus | 0.6022 | 0.4091 | 0.1301 | 0.5677 |
| 31 | Middle temporal gyrus | 0.5540 | 0.3661 | 0.0292 | 0.5175 |
| 32 | Inferior temporal gyrus | 0.5917 | 0.4069 | 0.0728 | 0.6038 |
| 33 | Temporal pole and planum polare | 0.5751 | 0.4081 | 0.0833 | 0.3690 |
| 34 | Transverse temporal gyrus | 0.8636 | 0.4912 | 0.3198 | 0.4469 |
| 35 | Planum temporale | 0.7830 | 0.5854 | 0.2568 | 0.7454 |
| 36 | Fusiform and parahippocampal gyri | 0.5435 | 0.3552 | 0 | 0.6082 |
| 37 | Hippocampus | 0.6045 | 0.2888 | 0.1546 | 0.5272 |
| 38 | Amygdala | 0.7984 | 0.3814 | 0.1291 | 0.5663 |

*portion of the precentral gyrus*, and the *postcentral gyrus including its medial segment*. Destrieux's pre- and post-central gyri only occupy the lateral brain surface, and it lists a single *paracentral lobule* on the medial surface. We therefore subdivided the paracentral lobule into an anterior (motor) and posterior (sensory) portion. In the premotor area, MNIA lists *medial frontal cortex*, *medial segment of the superior frontal gyrus* and *supplementary motor cortex*. Destrieux offers only the single possibility of *superior frontal gyrus* for all these. We therefore also subdivided the latter into three regions. The Supporting Information table lists the entire

**Table 2. Fractional state-related changes of normalized sample entropy.**

| | MNIA region | W-N2 | W-N3 | W-REM |
|---|---|---|---|---|
| 1 | Superior and middle occipital gyri | 0.2182 | 0.9182 | −0.1436 |
| 2 | Inferior occipital gyrus & occipital pole | 0.0681 | 0.8772 | −0.0534 |
| 3 | Cuneus | 0.3814 | 0.7043 | 0.1044 |
| 4 | Calcarine cortex | 0.4369 | 0.7853 | 0.4184 |
| 5 | Lingual gyrus & occipital fusiform gyrus | 0.3784 | 0.8295 | 0.2270 |
| 6 | Postcentral gyrus with medial segment | 0.4866 | 0.9205 | 0.1315 |
| 7 | Superior parietal lobule | 0.2710 | 0.8632 | −0.0338 |
| 8 | Parietal operculum | 0.3469 | 0.7919 | −0.1287 |
| 9 | Supramarginal gyrus | 0.3691 | 0.8648 | 0.0604 |
| 10 | Angular gyrus | 0.4007 | 0.9493 | 0.0088 |
| 11 | Precuneus | 0.2684 | 0.8108 | −0.0579 |
| 12 | Posterior cingulate | 0.5184 | 0.8701 | 0.2811 |
| 13 | Anterior insula | 0.4364 | 0.8904 | 0.0912 |
| 14 | Posterior insula | 0.4469 | 0.8597 | 0.1910 |
| 15 | Gyrus rectus and orbital gyri | 0.4798 | 0.9029 | −0.1478 |
| 16 | Anterior cingulate | 0.3983 | 0.8052 | 0.0160 |
| 17 | Middle cingulate | 0.3633 | 0.7002 | 0.0345 |
| 18 | Supplementary motor cortex | 0.3920 | 0.7512 | −0.0566 |
| 19 | Medial frontal cortex | 0.5524 | 0.9865 | 0.0251 |
| 20 | Central operculum | 0.4150 | 0.7890 | −0.0702 |
| 21 | Frontal operculum | 0.4236 | 0.8848 | 0.0304 |
| 22 | Opercular part of inferior frontal gyrus | 0.4397 | 0.7940 | 0.0974 |
| 23 | Triangular part of inferior frontal gyrus | 0.3929 | 0.7339 | −0.0863 |
| 24 | Orbital part of inferior frontal gyrus | 0.4197 | 0.8461 | −0.0277 |
| 25 | Middle frontal gyrus | 0.4561 | 0.7775 | −0.0260 |
| 26 | Superior frontal gyrus and frontal pole | 0.4798 | 0.8428 | −0.1155 |
| 27 | Medial segment of superior frontal gyrus | 0.5842 | 0.8210 | 0.0473 |
| 28 | Medial segment of precentral gyrus | 0.4971 | 0.8603 | 0.2394 |
| 29 | Precentral gyrus | 0.4064 | 0.7210 | 0.1216 |
| 30 | Superior temporal gyrus | 0.3207 | 0.7840 | 0.0573 |
| 31 | Middle temporal gyrus | 0.3392 | 0.9473 | 0.0659 |
| 32 | Inferior temporal gyrus | 0.3123 | 0.8770 | −0.0204 |
| 33 | Temporal pole and planum polare | 0.2904 | 0.8552 | 0.3584 |
| 34 | Transverse temporal gyrus | 0.4312 | 0.6297 | 0.4825 |
| 35 | Planum temporale | 0.2524 | 0.6720 | 0.0480 |
| 36 | Fusiform and parahippocampal gyri | 0.3465 | 1.0000 | −0.1190 |
| 37 | Hippocampus | 0.5222 | 0.7443 | 0.1279 |
| 38 | Amygdala | 0.5223 | 0.8383 | 0.2907 |

MNIA-Destrieux correspondence. The subdivisions of the paracentral lobule and superior frontal gyrus are readily visible on the medial cortical surface views of Figs 1 and 2.

## Results

The differences in the distribution of SE between the wake and N2, and wake and N3 were highly significant ($p < 10^{-12}$ for both comparisons), whereas the difference between the

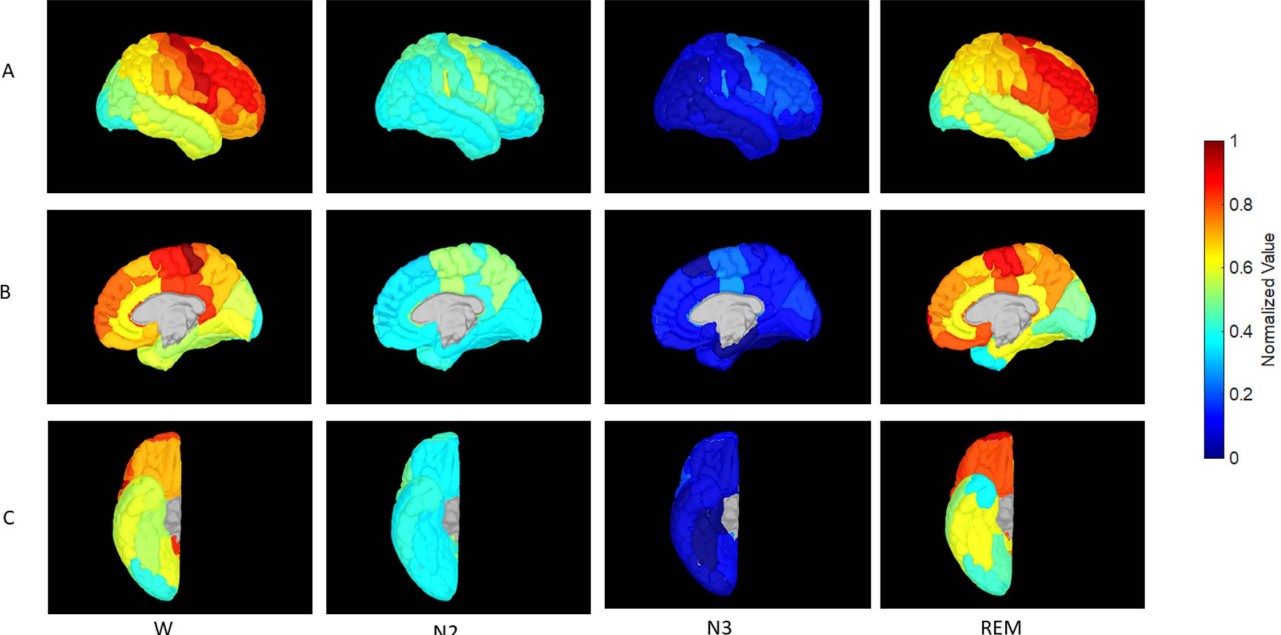

**Fig 1. Gyral views of the lateral cortical surface during the wake, N2 sleep, N3 sleep and REM sleep stages.** Normalized sample entropy values are color coded with cool and hot colors corresponding to the [0,1] range. Normalization was achieved by defining $SE_n = (SE - SE_{min})/(SE_{max} - SE_{min})$ where $SE_{max}$ was the maximum SE over all brain areas and conditions and $SE_{min}$ was the minimum. The wake state reveals an overall rostro-caudal hot-to-cold gradient. Superimposed is the sensorimotor gradient that decays away from the Rolandic cortex. Heschl's gyrus (seen in inflated view) and the planum temporale show a similar gradient. SE values decrease globally in N2 and N3 sleep, though the sensorimotor and auditory gradients are maintained. In REM sleep the SE of prefrontal regions exceed those of Rolandic cortex in a reversal of the wake relationship. The primary and posterior association auditory cortices and the primary and association visual areas behave similarly.

wake and REM states was not ( $p = 0.12$ ). Figs 1 and 2 show region-specific colored gyral and pial surface cortical maps of the SE profiles of brain regions during wakefulness, N2 sleep, N3 sleep, and REM sleep. Fig 3 shows histograms of the SE distribution for all four conditions.

Stability of spatial patterns of SE across sleep-wake states were also highly significant. In the usual notation and with the subscript $c$ denoting cosine similarity, the summary statistics of the distribution of CS of the shuffled SE vectors in the wake state W were

$$\mu_{cW} = 0.9688; \sigma_{cW} = 0.0047.$$

Similar figures for the sleep states were

$$\mu_{cN2} = 0.9746; \sigma_{cN2} = 0.0039;$$

$$\mu_{cN3} = 0.7614; \sigma_{cN3} = 0.0379;$$

$$\mu_{cREM} = 0.9527; \sigma_{cREM} = 0.008.$$

Thus, state vectors W, N2 and REM were intrinsically homogeneous (high mean CS with small variances); state N3 was less so. However, CS between states W and N2 was even larger:

$$CS_{W,N2} = 0.9892,$$

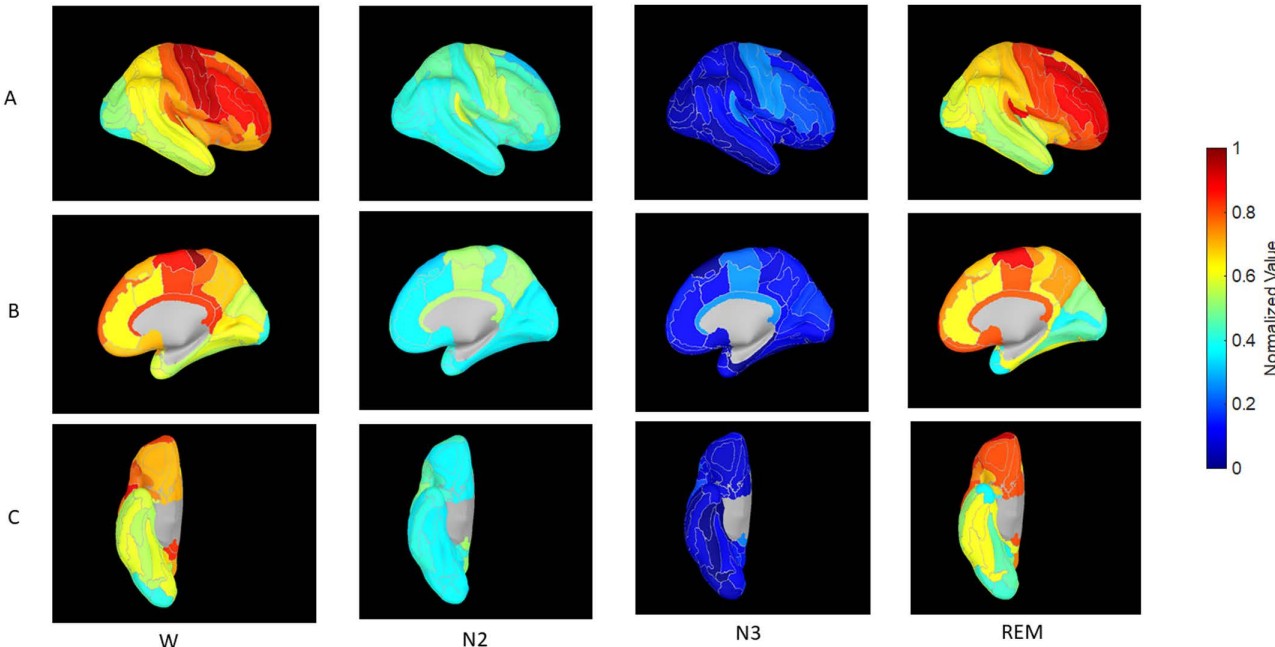

**Fig 2. Inflated cortical views corresponding to Figure 1.**

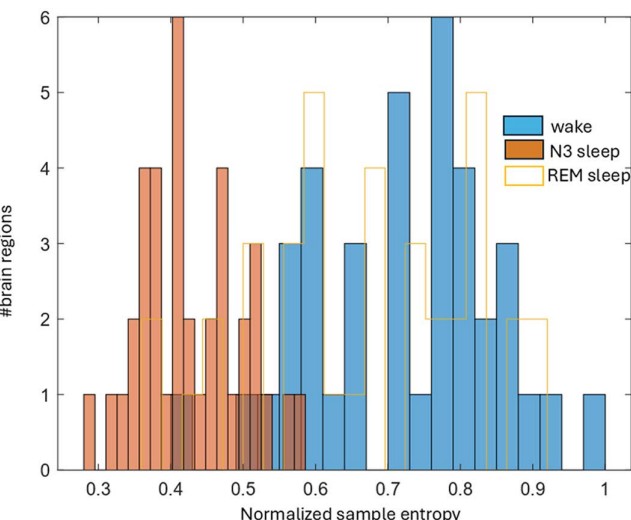

**Fig 3. Histogram of normalized sample entropy values for the wake (solid blue), N3 sleep (solid brown) and REM sleep (hollow yellow) states (first, third and fourth numerical columns of Table 1).** Values shift to a wake-like appearance in REM.

a highly significant difference from $\mu_{cN2}$ ($p < 10^{-4}$). The equivalent comparisons between N2 and N3, and wake and REM were

$$CS_{N2,N3} = 0.9109; \quad CS_{W,REM} = 0.9862.$$

$CS_{N2,N3}$ was significantly larger than $\mu_{cN3}$ and $CS_{W,REM} = 0.9862$ significantly larger than $\mu_{cREM}$ ($p < 10^{-4}$ for both comparisons).

SE in the wake state followed an RCG in an approximate overall manner, with high values anteriorly and a global minimum in the posterior cortex. Primary motor cortex had the highest SE of all: it was indeed the global maximum. This SE maximum declined over premotor frontal lobe both medially and laterally (with the highest values in this group over the opercular inferior frontal gyrus, followed by the middle frontal gyrus (MFG)), as well as the mid- and posterior cingulate regions. A comparably high value to the MFG was also seen in primary auditory cortex over Heschl's gyrus with lower values in the more posterior auditory areas (planum temporale). Sensory areas and the mid and posterior cingulate came in next: primary sensory cortex had higher SE than sensory association areas. The majority of the temporal neocortex and parietal lobe had values in the high mid-range values as did the amygdala, followed by the hippocampus. A global minimum was achieved in the extreme posterior cortex – occipital pole – with slightly higher values over the remainder of the occipital lobe and cuneus. SE was higher over pericalcarine cortex than over proximate visual association areas. Sleep N2 was accompanied by a decrease in SE in all brain regions except for the occipital pole and inferior occipital gyrus. Sleep N3 appeared to be an accentuated version of N2, with all SE numbers falling further, and the occipital pole and inferior occipital gyrus no longer holding out near-wake values. In REM the precentral gyrus lost its primacy to the prefrontal cortex which became a large swathe of SE maximum that exceeded waking values. Heschl's gyrus retained SE value of N2 but planum temporale recovered to near-wake values. The situation was analogous in the occipital cortex: The primary visual area (calcarine cortex) retained low N2-like values, but the remainder of the occipital lobe recovered (or exceeded) the SE values attained in wakefulness. Most of the temporal neocortex and parietal lobe returned to values comparable to the waking state. Thus, most of the brain 'woke up' during REM (fractional wake-REM values are near zero or negative: column 3, Table 2), but relative exceptions occurred over the primary sensory and motor cortex - calcarine cortex, the pre- and post-central gyri and transverse temporal gyrus - as well as over the lingual and occipital fusiform gyri, the posterior insula, the temporal pole and planum polare, and the hippocampus and amygdala.

## Discussion

This work was motivated by the larger question of why brain rhythms appear the way they do over different regions of the cortex. We used the MNIA intracerebral EEG (iEEG) atlas as the data source and sample entropy (SE) as a complexity metric to explore the entire cortical surface and the amygdala-hippocampus. We were interested in whether the spatial changes of SE had an identifiable pattern, and whether those bore a relation to systematic variations in local tissue-level properties of the cerebral cortex. An important example of the latter is the whole-brain rostro-caudal gradient (RCG) in neuronal density [10]. Superimposed on RCG are other gradients radiating away from primary somatic sensorimotor, visual and auditory regions to association areas, in cortical thickness [11], intracortical myelin [12] and intrinsic neuronal timescale (INT) [13, 14]. The RCG is thought to relate to the differing lengths of neurogenesis during development, with neurogenesis persisting for longer in the caudal brain, this permitting a greater number of cell cycles [15]. Sensorimotor-to-transmodal gradients are understood as spatial templates that reflect information processing hierarchies and thus comprise an intrinsic coordinate system for the cerebral cortex [12].

The science of complexity – famously birthed in Weaver's classic essay [16] - studies entities comprising large numbers of agents that interact relatively simply. By this characterization, the central nervous system must qualify as a complex system *par excellence*, whether at

a whole-brain level, or at the level of neuronal populations sensed by a recording clinical electrode. Here we simply documented the distribution of a complexity metric (SE) of electrode activity (i.e., the intracerebral EEG) over the cortical surface. Preliminarily, we mention that simple moment-based statistics (signal standard deviation or RMS power) that only measure amplitude deviation from the mean signal are insensitive to finer variations, and related spectral methods (e.g., autocorrelation and signal bandwidth) may not be sensitive enough to diagnose subtle irregularities. Nonlinear techniques such as Lyapunov exponents, correlation dimension and Kolmogorov-Sinai (K-S) entropy were designed for deterministic dynamical systems, and in practice require large amounts of data for their computation. Approximate entropy (AE) was introduced to address exactly the requirements for a metric that was model-free and applicable to short noisy datasets. Sample entropy was subsequently introduced as a refinement to AE, to correct some of the bias inherent in AE. Since then, SE has been used in hundreds of studies involving diverse biomedical datasets. We are aware of a further refinement of SE – fuzzy entropy (FE) [17] – that is claimed to offer superior results still. However, the literature on FE for biomedical datasets is currently scant, and we chose to go with the better-established SE. A different reason for our choice of SE was simply to achieve consistency with our prior work that involved SE. In the definition of SE, the parameter $m$ represents the window of time that is used as a comparator. $m = 2$ thus represents a narrow window (three points long) and therefore picks up fine time-dependent variations in the signal. Having $m$ larger elongates the time window and coarsens resolution. The parameter $r$ is the fractional standard deviation (thus, its values are constrained $0 < r < 1$). Values approaching unity allows for large variation in successive windows and would be insensitive to finer variations, and values approaching zero would be oversensitive and pick up the noise floor. Finally, trials of combinations of small $r$ and large $m$ did not yield computable values of SE for our data. Thus, we consider our choices of $m = 2$, $r = 0.2$ appropriate, and these same values have been ratified in the literature.

Our choice of SE was nevertheless one several alternative metrics that have been used for documenting the complexity of biomedical time series. We are aware of three recent publications that perform a similar analysis to ours with different metrics (Higuchi fractal dimension and intrinsic neuronal timescale (INT)) on the same dataset [18–20]. The relationship of SE to INT was particularly intuitive: high SE signals may be expected to exhibit short autocorrelations, and thus short INTs. The opposite would be true for low SE signals. However, our work here, related to prior observations of SE over the frontal lobe [21], carries an interpretation based on brain systems identified by cortical gradients rather than numbered brain areas.

We found an exact correspondence between the spatial behavior of the above structural metrics and the SE of resting intracerebral EEG. Specifically, we observed an overall RCG, with high values of SE anteriorly and the lowest waking SE over the inferior occipital gyrus and occipital pole (Table 1). We also observed gradients from primary to association areas in the motor, sensory, auditory and visual systems superimposed on the overall RCG. In the waking state the highest SE of all was confirmed over primary motor cortex (M1), a question we had posed to motivate this study. In the auditory system, the primary auditory area – the transverse temporal gyrus – exhibited large waking SE, with polymodal association areas (parietal lobe and temporal neocortex) exhibiting lesser, intermediate values. Similarly, primary sensory cortex had higher SE than sensory association areas. The lowest waking SE values occur over the primary visual and visual association areas in keeping with the global RCG, but even here SE was higher over primary, compared to visual association areas.

The descent to slow wave sleep was accompanied by a drop in SEs in all brain regions to more than 70% of their waking values (Table 2). That is, the histogram for SE distribution in the wake state shifts to the left in NREM sleep, and as expected, shifts back to the right

in REM (Fig 3). However, these sleep-related changes are not just left-right shifts; the N3 histogram also has narrower range. In other words, the spread of SE values – cortical gradients - seen in wakefulness shrinks. Gradients are thought to reflect information processing hierarchies – primary areas of high SE (fast INTs) projecting to association areas of low SE (slower INTs). In the setting of reduced cortical information processing (e.g., sleep) one may expect gradients to correspondingly reduce. These observations are concordant with a body of work examining information transmission within the cortex in the wake and sleep states. In NREM sleep the transmission of information fragments [22], an observation that is thought to underlie the evident inability of the brain to process complex information as well as consciousness itself [23].

The most intriguing changes were however in REM, where Table 2 (Column 3) shows that most brain areas returned to near their waking state SE (fractional changes near zero or negative). However, the primary visual and auditory areas did not: both calcarine cortex and transverse temporal gyrus showed large fractional wake-REM change, thus remaining relatively 'asleep'. This effect was also present though less pronounced in primary motor (M1) and sensory (S1) areas, the posterior cingulate, the posterior insula, the cuneus and occipital fusiform gyrus, and the amygdala-hippocampus. These are similar findings to our prior spectral modelling study of the MNIA data [9], and for the visual cortex, the study of Gorgoni and co-workers [24]. The differential REM reactivation of primary and association cortex presumably reflects the neural correlates of sensory experience in the absence of the elemental sensations themselves.

In summary, we demonstrate that the normal human intracranial EEG exhibits rich spatial structure in the distribution of SE. Our results deepen the conventional interpretation of iEEG from its frequency content attributes (Berger bands, rhythmic activity, etc.) to a spatial perspective related to cortical biology. In particular, we show that SE tracks temporal processing hierarchies in cerebral cortex, and in a concordant fashion to the distribution of other variables that reflect cortical structure. However, the spatial architecture of SE over the cortex is highly sleep-wake dependent; little remains of the awake spatial gradient in N3 sleep, for instance. Further exploration of this state-dependent dissociation between structural and functional cortical gradients is of interest. The alteration of iEEG gradients by pathology (e.g., epileptic foci) also remains an area for further work.

## Supporting information

**S1 Table. Mapping of the 38 MNIA brain regions to the 74 anatomical parcellations in the Destrieux atlas.** In general, the larger numbers of Destrieux brain regions led to several single MNIA regions mapping to multiple Destrieux brain regions. Over the frontomedial brain surface the MNIA parcellation is finer than Destrieux and required subdivision of Destrieux's paracentral lobule and superior frontal gyrus into three sub-regions each (see text).
(DOCX)

## Author contributions

**Conceptualization:** Giridhar Kalamangalam.

**Formal analysis:** Giridhar Kalamangalam, Ioan Mircea Chelaru, Abbas Babajani-Feremi.

**Funding acquisition:** Giridhar Kalamangalam.

**Writing – original draft:** Giridhar Kalamangalam.

**Writing – review & editing:** Ioan Mircea Chelaru, Abbas Babajani-Feremi.

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
