## [Decision Letter · Decision Letter 0]

8 Oct 2024

PONE-D-24-18203GRADIENTS IN SIGNAL COMPLEXITY OF SLEEP-WAKE INTRACEREBRAL EEGPLOS ONE

Dear Dr. Kalamangalam,

Thank you for submitting your manuscript to PLOS ONE. After careful consideration, we feel that it has merit but does not fully meet PLOS ONE’s publication criteria as it currently stands. Therefore, we invite you to submit a revised version of the manuscript that addresses the points raised during the review process.

Reviewers  highlighted a number of weakinesses that must be addressed. Concerns include clarification of study design, statistical treatment, and overall polishing of the mauscript, that looks unrefined/uderdeveloped in several places.

Note that some of the concerns require substantial effort on your part to be addressed. New processing and a major rewriting are expected I encourage the authors to fully address the points highlighted by reviewers #2 and #3.

We look forward to receiving your revised manuscript.

Kind regards,

Federico Giove, PhD

Academic Editor

PLOS ONE

Reviewers' comments:

Reviewer's Responses to Questions

**Comments to the Author**

1. Is the manuscript technically sound, and do the data support the conclusions?

Reviewer #1: Yes

Reviewer #2: Partly

Reviewer #3: Partly

2. Has the statistical analysis been performed appropriately and rigorously? 

Reviewer #1: Yes

Reviewer #2: No

Reviewer #3: No

3. Have the authors made all data underlying the findings in their manuscript fully available?

Reviewer #1: Yes

Reviewer #2: Yes

Reviewer #3: Yes

4. Is the manuscript presented in an intelligible fashion and written in standard English?

Reviewer #1: Yes

Reviewer #2: Yes

Reviewer #3: Yes

5. Review Comments to the Author

Reviewer #1: The paper presents a comprehensive and insightful analysis of Sleep-Wake Intracerebral EEG, demonstrating a strong grasp of both the theoretical and practical aspects of EEG analysis. The idea is robust, and the results are clearly articulated, offering valuable contributions to the field of neuroscience. The discussion effectively contextualizes the findings within existing literature, highlighting the paper's significance and potential impact on future research.

Reviewer #2: The authors set out to investigate the complexity of neural responses in four different arousal states (i.e., wakefulness, N2, N3, and REM) using established cortical recordings. A spatial distribution estimated using sample entropy (SE) was obtained for each of the four conditions. A similar response localization (i.e., RCG) was found in wakefulness and REM, which differed from those detected in the other two conditions (i.e., N2 and N3). Based on these findings, the authors concluded that cortical activity exhibited a rich spatial structure (e.g., line 303), and a brief discussion of the association with previous works was provided. To meet the journal’s standard for publication as a research article, several aspects require further investigation and clarification.

First, why was SE chosen as a neural readout? As far as I know, many different types of neural measurements can reflect signal complexity, and several were also mentioned by the authors in the manuscript (e.g., lines 247 to 249). The reason for preferring SE over others should be made clearer. A comparison between the spatial patterns reflected by SE and those reflected by other readouts is necessary to demonstrate the distinctiveness of SE. Additionally, the parameters (e.g., m and r) used in the estimations require further clarification. While following convention (line 138) is a good approach in many cases, an explicit rationale for doing so should be specified. For instance, is it possible that the spatial pattern would still occur even if a different set of parameters were selected?

Second, the conclusions were drawn based on observations, but statistical analysis is needed to support them. To eliminate the possibility that the identified spatial patterns were a random effect, permutations or similarity measures could be used to demonstrate the stability of the effect. For instance, one could shuffle the SE values within a condition (e.g., shuffle SE in space) several times, and then use vectorized similarity measures (e.g., cosine similarity) to estimate the spatial dissimilarity of the randomizations across conditions to see whether the real spatial effect surpassed a threshold. Alternatively, representational similarity analysis (RSA) could be used to directly compare the cortical distribution among conditions.

Third, from my experience, the article is somewhat brief, and the dataset appears under-analyzed compared to standard practice, especially when compared to studies using direct cortical recordings. The brevity and inadequate analyses make it difficult to draw strong conclusions, either theoretically or from a data-driven perspective. For example, it is hard to determine whether the spatial effect was driven by the different arousal states or the chosen neural measurements. It is possible that the spatial patterns would have disappeared if different parameters or neural measurements had been used.

Fourth, the figures shown in the manuscript lack a scale indicator, i.e., the correspondence between the colors in the figures and the numbers reflecting the degree of activation. Although we can infer, empirically, that the colors from blue to red reflect a range from low (0) to high (1) in response levels, an explicit colormap is necessary.

Lastly, the writing is somewhat repetitive across sections. For instance, the end of the methods section (from line 145 onward) and the main results (e.g., Figures 1 and 2) describe similar results using different approaches. Instead of describing the spatial effects (i.e., where the activations occurred), more space should be dedicated to discussing the implications of the results and their association with previous studies in a logical order.

Reviewer #3: In this paper, the authors aimed to understand the relationship between cortical signals recorded using iEEG and the underlying cortical architecture. They used Sample Entropy (SE) as the primary metric to examine signal complexity across the entire cortical surface. The authors found that signal complexity follows a rostrocaudal gradient during wakefulness, with higher SE values in anterior regions, particularly in the primary motor area, and a global minimum in the posterior cortex. They then analyzed changes in SE between wakefulness and different sleep stages, finding that SE decreased in stages 2 and 3, while SE in REM sleep remained largely similar to the wake state. The authors conclude that signal complexity reflects the processing hierarchy and other structural and functional characteristics of the cortex.

I have a few major comments:

1. None of the observations are supported by any statistical testing. No p-values and sample sizes have been reported. Because of this, the conclusions feel largely speculative. Supporting the conclusions with proper statistical validation would make the findings more convincing.

2. SE is one way to measure signal complexity, and its definition does not allow much insights into the neural dynamics. Additionally, SE is sensitive to hyperparameter selection. How would these gradients change if m and r were adjusted? Can the authors provide a rationale for setting r at 0.2?

3. For more rigorous conclusions, it would be beneficial to explore other complexity metrics that provide more detailed insights into the nature of neural dynamics.

4. The paper could benefit from clearer structure. Currently, key conclusions are mixed with observations, making it challenging to follow the main arguments.

6. Lines 98-100 describe the main question of the study, but the text does not clearly provide an answer. A more direct connection between the results and the research question would improve clarity.

7. How were the SE values normalized across sleep stages to allow for appropriate comparisons?

8. Minor comment: Please add color bars to all figures.

6. PLOS authors have the option to publish the peer review history of their article (what does this mean? ). If published, this will include your full peer review and any attached files.

**Do you want your identity to be public for this peer review?** For information about this choice, including consent withdrawal, please see our Privacy Policy .

Reviewer #1: **Yes: ** Supriya

Reviewer #2: No

Reviewer #3: No

---

## [Author Response · Author response to Decision Letter 1]

16 Dec 2024

All editorial and reviewer comments have been addressed. Funding information has been added.

---

## [Editor Report · Decision Letter 1]

28 Jan 2025

PONE-D-24-18203R1GRADIENTS IN SIGNAL COMPLEXITY OF SLEEP-WAKE INTRACEREBRAL EEGPLOS ONE

Dear Dr. Kalamangalam,

Thank you for submitting your manuscript to PLOS ONE. After careful consideration, we feel that it has merit but does not fully meet PLOS ONE’s publication criteria as it currently stands. Therefore, we invite you to submit a revised version of the manuscript that addresses the points raised during the review process.

Unfortunately, previous reviewers were not available to review this revised version.

The authors made a substantial effort to address comments and criticism by reviewers, and I think that the manuscript is improved.

I found two residual problems

Point 2 by reviewer #2 (Also raised by reviewer #3). Authors should introduce in the main text the reasoning and literature behind their choice. It is not needed that figures for review are introduced in the manuscript, but the rationale should.I think that the statistical analysis should be extended to the spatial patterns, as suggested by reviewer #2

We look forward to receiving your revised manuscript.

Kind regards,

Federico Giove, PhD

Academic Editor

PLOS ONE
---

## [Author Response · Author response to Decision Letter 2]

20 Feb 2025

We thank the reviewer further comments on our manuscript. The concerns raised are reproduced in italicized font below, and our response follows immediately beneath.

The authors made a substantial effort to address comments and criticism by reviewers, and I think that the manuscript is improved.

Response: Thank you.

Point 2 by reviewer #2 (Also raised by reviewer #3). Authors should introduce in the main text the reasoning and literature behind their choice. It is not needed that figures for review are introduced in the manuscript, but the rationale should.

Response: In addition to explanatory text in the Discussion in our prior revision, we have now added one more citation and some edits. The second paragraph in the Discussion section describes the rationale for our choice of SE as a metric of signal complexity and the justification for our choice of parameters.

I think that the statistical analysis should be extended to the spatial patterns, as suggested by reviewer #2.

Response: Our prior statistical analysis was restricted to signed rank-sum comparison of each of the three sleep states (N2, N3 and REM) to the wake state (W). We accept that extension of statistical analysis to spatial considerations will strengthen the manuscript. We have now done so. The methods are a prior reviewer’s suggestion (for which we give our thanks) of using shuffled distributions within a single condition, observing cosine similarities of the shuffled variants, and comparing these across the sleep-wake states. This analysis is mentioned in the Methods, and the results appear under Results.

Finally, we also took the opportunity to make some additional minor stylistic edits to the manuscript.

---

## [Editor Report · Decision Letter 2]

24 Feb 2025

GRADIENTS IN SIGNAL COMPLEXITY OF SLEEP-WAKE INTRACEREBRAL EEG

PONE-D-24-18203R2

Dear Dr. Kalamangalam,

We’re pleased to inform you that your manuscript has been judged scientifically suitable for publication and will be formally accepted for publication once it meets all outstanding technical requirements.

Kind regards,

Federico Giove, PhD

Academic Editor

PLOS ONE
---

## [Editor Report · Acceptance letter]

PONE-D-24-18203R2

PLOS ONE

Dear Dr. Kalamangalam,

I'm pleased to inform you that your manuscript has been deemed suitable for publication in PLOS ONE. Congratulations! Your manuscript is now being handed over to our production team.

Kind regards,

on behalf of

Dr. Federico Giove

Academic Editor

PLOS ONE